# Variations in Microstructural and Physicochemical Properties of Candelilla Wax/Rice Bran Oil–Derived Oleogels Using Sunflower Lecithin and Soya Lecithin

**DOI:** 10.3390/gels7040226

**Published:** 2021-11-22

**Authors:** Deblu Sahu, Deepti Bharti, Doman Kim, Preetam Sarkar, Kunal Pal

**Affiliations:** 1Department of Biotechnology and Medical Engineering, National Institute of Technology, Rourkela 769008, Odisha, India; 519BM1005@nitrkl.ac.in (D.S.); 517BM1004@nitrkl.ac.in (D.B.); 2Department of International Agricultural Technology & Institute of Green BioScience and Technology, Seoul National University, Seoul 151747, Gwangwon-do, Korea; kimdm@snu.ac.kr; 3Department of Food Process Engineering, National Institute of Technology, Rourkela 769008, Odisha, India; sarkarpreetam@nitrkl.ac.in

**Keywords:** candelilla wax, lecithin, oleogels, color parameters, spreadability

## Abstract

Candelilla wax (CW) is a well-known oleogelator that displays tremendous oil-structuring potential. Lecithin acts as a crystal modifier due to its potential to alter the shape and size of the fat crystals by interacting with the wax molecules. The proposed work is an attempt to understand the impact of differently sourced lecithin, such as sunflower lecithin (SFL) and soya lecithin (SYL), on the various physicochemical properties of CW and rice bran oil (RBO) oleogels. The yellowish-white appearance of all samples and other effects of lecithin on the appearance of oleogels were initially quantified by using CIELab color parameters. The microstructural visualization confirmed grainy and globular fat structures of varied size, density, packing, and brightness. Samples made by using 5 mg of SFL (Sf5) and 1 mg of SYL (Sy1) in 20 g showed bright micrographs consisting of fat structures with better packing that might have been due to the improvised crystallinity in the said samples. The FTIR spectra of the prepared samples displayed no significant differences in the molecular interactions among the samples. Additionally, the slow crystallization kinetics of Sf5 and Sy1 correlated with better crystal packing and fewer crystal defects. The DSC endotherm displayed two peaks for melting corresponding to the melting of different molecular components of CW. However, all the formulations showed a characteristic crystallization peak at ~40 °C. The structural reorganization and crystal growth due to the addition of lecithin affected its mechanical property significantly. The spreadability test among all prepared oleogels showed better spreadable properties for Sf5 and Sy1 oleogel. The inclusion of lecithin in oleogels has demonstrated an enhancement in oleogel properties that allows them to be included in various food products.

## 1. Introduction

In the past few years, a large human population has inclined towards healthier and nutrition-rich food products. Diets that include an excessive amount of trans fatty acids (TFAs) and saturated fatty acids (SFAs) lead to ailments such as hypertension, obesity, and cardiovascular diseases [1]. However, several studies reported that replacing saturated fat with unsaturated fat shows a positive health impact, even partially in daily life. Conferring to American Heart Association guidelines reduced intake of high SFAs and TFAs content diet and increased unsaturated fatty acids (USFAs). These are usually found in vegetable oils (e.g., sunflower oil, rice bran oil, etc.), highly recommended for better cardiac health [2,3]. Among the USFAs, poly-unsaturated fatty acids (PUFAs) are highly enriched in oils. In this regard, microalgae have gained attention as a probable source of ω-3 fatty acids [4]. Following the increasing consumer health-related concern and guidelines of regulatory bodies, many food industries and researchers around the globe have focused on developing new techniques or products with equivalent functionality. From this perspective, oleogels that show excellent oil-structuring properties could better replace the traditional trans- and saturated-fat structuring methods. Oleogels contain vegetable oils that are free from trans fats and contain fewer SFAs than “traditional” solid fats (e.g., butter, tallow, margarine, etc.). Their use as ingredients might improve the nutritional profile of the food products.

Oleogels are semisolid formulations composed of a higher amount of liquid (usually above 90% *w*/*w* oil) and a deficient concentration of solids (known as gelators). Ideally, to be used as oleogelators in foods, materials should possess properties such as (a) a natural origin, (b) Generally Recognized as Safe (GRAS) status, (c) presence of an interacting and a lipophilic part, (d) self-assembling properties, and (e) thermoreversible properties [5,6]. Examples of oleogelators are natural waxes [7,8], ethylcellulose [9], monoacylglycerols [10,11], long-chain fatty acids, and hydroxyl fatty acids [12]. Among all of these oleogelators, natural waxes are more efficient because they can form a fat crystal network with strong oil-binding properties, even at low concentrations (less than 10% *w*/*w*) [13]. Natural waxes are extracted from seed coating (e.g., rice bran wax), plant cuticles (e.g., carnauba wax), or the secretion of an insect (e.g., beeswax). Usually, natural waxes have diverse chemical compositions, including a long chain of many hydrophobic substances such as fatty alcohols, hydrocarbons, free fatty acid, ketones, mono-, di-, tri-acylglycerols, and other minor components [14]. The wax melts in oil upon heating, and they crystallize upon cooling by developing a thermoreversible network of wax crystals [15]. Waxes with different chemical compositions exhibit different melting points, altering their gelation and crystallization behaviors in oils [16]. To date, several studies on oleogels have been made with candelilla wax (CW), sunflower wax (SW), rice bran wax (RW), paraffin wax (PW), beeswax (BW), and carnauba wax (CRW) [17,18,19,20]. However, in recent years, plant-based waxes (e.g., CW, SW, RW, CRW, etc.) became quite influential in the field of oleogel research. The possible reason behind this is that their excellent gelation properties, oil-binding capacity, and easy availability [21].

CW is an FDA-approved food additive that is garnered from the leaves of small shrubs (*Euphorbia antisyphilitica* and *Euphorbia cerifera*). These shrubs are found in explicit parts of Mexico and Texas, USA. This natural wax has mainly found applications as a binder in certain food products (e.g., chewing gum), glazing agents, and pharmaceutical products [22]. CW comprises 50–65% of high melting alkanes (with 24–34 carbons), 27–35% esters of acids and alcohols, 7–10% free fatty acids, and 10–15% free fatty acids alcohols [23]. However, oleogels prepared with a single gelator had offered less tunable gel properties. As a solution to the limitation mentioned above, scientists are exploring multicomponent gel systems that offer flexibility to alter and optimize various gel parameters, such as thermal properties, storage stability, and mechanical behavior. This may further enhance its applicability in different food products [24,25]. One of the components of such a system is lecithin, which is an emulsifier that acts as a crystal habit (size) modifier. Lecithin is a natural amphiphilic phospholipid that is generally extracted from the biological membranes of plant cells. It is a composite mixture of phosphatidylcholine, phosphatidylethanolamine, phosphatidylserine, and phosphatidylinositol, having some amounts of triglycerides and fatty acids [26]. The interaction of lecithin with wax molecules alters the morphology and dimension of the resultant fat crystals [27]. Apart from its natural origin, the other motivation for using lecithin in food products is its ability to self-assemble through non-covalent interactions, usually hydrogen bonding [28]. In oleogels, lecithin is used to tailor the crystal morphology of the gelators, which is accomplished by tuning the lecithin concentration [27].

Our thorough survey of the literature indicates that the formation and gelation behavior of rice bran oil (RBO) and CW-based oleogels in the presence of lecithin have not been addressed yet. The present work aims to prepare oleogels by mixing differently sourced lecithin in the presence of RBO and CW (Section 4.2). Two types of lecithin, namely soya lecithin (SYL) and sunflower lecithin (SFL), were used in varying concentrations to tailor the crystal morphology of CW in the oleogels [29]. The differently sourced lecithins also vary from each other in terms of their composition. For example, the phospholipid composition in SYL consists of 15% phosphatidylcholine (PC), 10% phosphatidylinositol (PI), 11% phosphatidylethanolamine (PE), and 4% phosphatidic acid (PA). SFL comprises 16% PC, 8% PE, 14% PI, and 3% PA. Similarly, there exists a difference in the fatty acid composition of both the lecithin as well. SYL consist of 55% of linoleic acids, 17% oleic acids, 16% palmitic acids, and 7% α-linolenic acids of the total fatty acids. However, in SFL, of the total fatty acids, 63% are linoleic acids, 18% are oleic acids, and 11% are palmitic acids [30]. RBO was selected as the oil phase in the proposed multi-component gel system. The selection was based on low SFA content (~20% *w*/*w*). It also had higher nutritional components, such as antioxidants, tocotrienols, oryzanol, and plant sterols [30]. The thermal properties were examined through differential scanning calorimetry (DSC). A separate crystallization study was also performed at a fixed temperature. The texture properties of oleogels were elucidated through the spreadability test. The microstructure and color parameters were characterized with polarized microscopy and colorimetry analysis, respectively. Finally, the interaction of different components was investigated with Fourier-Transform Infrared (FTIR) spectroscopy.

## 2. Results and Discussion

### 2.1. Colorimetry Studies

The CIELab color parameters, i.e., L*, a*, and b* of prepared oleogels, were calculated. The lightness of luminance (L*) ranges from 0 (black) to 100 (white), whereas chromaticity parameters (a*, and b*) ranges from −120 to +120. The a* component varies from green (−ve) to red (+ve) while b* component varies from blue (−ve) to yellow (+ve) [31]. Studying color parameters is crucial as this is one of the desired sensory characteristics required for consumers’ acceptance. It was detected that the average values of L* for all samples fall in the range of 52 to 62 (Table 1), which indicates the samples are lighter in color [32]. A high L* value of food additive is desirable, as it does not interfere with the color perception of the food products [33]. The L* value obtained for the control sample was 57.45 ± 1.92. The inclusion of SFL in Sf1 showed no significant change from control in the L* value (57.71 ± 1.35). Further addition of SFL caused significantly lower L* values in Sf3 (52.76 ± 0.85) than Sf1 and the control (*p* < 0.05). Sample Sf5 shared similar luminance with Sf1 and the control; however, a significant increase in the L* values of Sf5 (55.89 ± 0.40) from Sf3 was observed. Sf10, where the SFL content was higher, the L* values similar to the control were observed. The addition of SYL in Sy1 and Sy3 did not influence the luminance compared to the control (*p* > 0.05). However, Sy5 (61.60 ± 0.36) and Sy10 (59.69 ± 0.82) displayed a significant increase in the L* value to the control (*p* < 0.05). Among SYL, all samples appeared similar in their luminance except Sy10, with a significant decrease in the L* value, as compared to Sy5 (*p* < 0.05). Interestingly, the L* value of Sy10 was not different from the L* values of the other SYL-added samples.

The a* value was found to be negative for all the samples. This suggested the presence of a green tone in the prepared samples. In control, a* value was observed to be −9.27 ± 3.13. The inclusion of SFL and SYL had no impact on the values of a* as these values were not significantly different from control. Among the SFL samples, the a* value in Sf3 and Sf5 was decreased considerably compared to the a* value of Sf1 (*p* < 0.05). Further, the green tone of Sf3 was similar to Sf5. At the highest SFL content in Sf10, there was a significant increase (*p* > 0.05) in the a* value from Sf5. Between the SYL samples, the a* values of Sy3 and Sy10 were statistically substantial from Sy1 (*p* < 0.05). There were no other significant differences observed among the SYL samples. All the oleogels showed a yellow tone, as evident from the positive b* values (Table 1). The b* value of the control was observed to be 37.14 ± 1.98. The changes in the b* value with the addition of SFL were found to be composition-dependent. Among the SFL-containing samples, Sf1 showed b* value that was similar to that of the control (*p* > 0.05). A consequent increase in the SFL content increased the b* value of Sf3 (*p* < 0.05), which was even higher than the control (*p* < 0.05). A further rise in the SFL content in Sf5 did not show any significant difference in the yellowness compared to the control, Sf1, and Sf3. Although Sf10, where the SFL content was highest, had yellowness, it was similar to that of Sf5. There was a marked reduction in its b* value from control, Sf1, and Sf3 (*p* < 0.05). The inclusion of SYL in the samples showed a similar yellowness in all samples, similar to the control, except for a significant decrease in the yellow tone of Sy10 from the control (*p* < 0.05). There was a reduction of the mean b* value of Sy3 as compared to Sy1; however, the drop was not statistically significant (*p* > 0.05). An increase in the SYL content showed a consequent reduction in the b* value of Sy5 from Sy1 and Sy3 that was statistically significant (*p* < 0.05). In Sy10, where the maximum SYL content, the b* value substantially lowered from the other SYL-containing samples.

The visual appearance of all the samples was yellowish-white. The samples’ different color parameters (e.g., Yellowness Index (YI) and Whiteness Index (WI)) were calculated. The YI was calculated from the L* and b* values. The YI of the control was 92.35 ± 4.49 (Table 1). The addition of SFL in Sf1 showed a similar YI to the control (*p* > 0.05). On increasing the SFL content, a significantly increased YI value of Sf3 (*p* < 0.05) was observed, which was even higher than control (*p* < 0.05). A further rise of lecithin in Sf5 resulted in a marked reduction of the yellowness from Sf3. However, Sf5 appeared similar to the control and Sf1 in terms of the YI value. At the highest SFL content, the Sf10 displayed the lowest YI value, and this lowering in yellowness was found substantial from the rest of the SFL-containing samples and the control. Among the SYL samples, Sy1 and Sy3 showed YI values similar to the control. The inclusion of SYL in Sy5 showed a significant reduction in the yellowness compared to Sy1, Sy3, and the control (*p* < 0.05). Although Sy10 shared similarities in the yellowness index to Sy5, the decreased value of YI in Sy10 was significant compared to Sy1 and Sy3. This reduced the YI value in Sy10 was also lesser than control (*p* < 0.05). Another color parameter that was calculated in the study was WI. WI is the degree of whiteness that combines L*, a*, and b* into a single term [34]. The values of WI observed in the study ranges from 40 to 50, which specifies the lightness of samples [17]. The WI of the control was calculated to be 42.69 ± 0.74 (Table 1). The SFL addition in the samples showed no effect in the whiteness of Sf1 from control. A further rise of the emulsifier content in sample Sf3 showed a significant drop in the value of WI from Sf1 and the control. In Sf5, the whiteness appeared similar to the control and Sf1; however, the increment in the WI value of Sf5 from Sf3 was significant (*p* < 0.05). A further rise of the SFL content to a maximum in Sf10 showed the highest WI value among SFL samples. This value was also higher than that of the control. The addition of SYL showed similar WI in Sy1 and Sy3 to that of the control. However, there was a significant increase of WI in Sy5 and Sy10 from the control (*p* < 0.05). The absolute color difference is obtained as a numerical value and is generally used to compare the samples with a particular standard. The control sample was considered as a standard in this study for carrying out this analysis. As reported in Reference [35], if the value of ΔE was greater than 3, the human eye can perceive the color difference. The calculated ΔE value in all the samples ranged from 4 to 10, suggesting that the color difference of the samples can be perceived easily by the human eye. The color parameters obtained in the study can be related to the microstructure.

### 2.2. Microscopy Studies

#### 2.2.1. Surface Topology

The surface topography of the prepared oleogels is depicted in Figure 1. The topography of the oleogels presented regularly distributed globular structures all over the matrices. The control sample showed uniformly distributed globular structures, which appeared as aggregates. The occurrence of these globular structures may be related to the crystalline domains of the fats. As visualized, the topography of emulsifier-containing oleogels demonstrated changes in the structural organization at different lecithin content. The incorporation of SFL in Sf1 and Sf3 resulted in bigger globular structures compared to the control. The density of these globular structures in Sf1 was observed to be almost similar to that of the control. However, in Sf3, a marginal reduction in density was observed. This might be due to the occurrence of more distinct spaces (yellow arrow) in its topography. However, the density of these globular structures was reduced till Sf3. A subsequent increase in SFL content in Sf5 resulted in highly dense globular structures, which was the most significant size amongst all the SFL-containing oleogels. This can be explained by the ability of lecithin to promote the formation of crystal aggregates in the lipid systems. The crystal appearance is modulated by the interactions of phosphatides of lecithin with triacylglycerols (TAGs) and free fatty acids. Accordingly, we observed that at lower content of SFL (i.e., 1, 3, and 5 mg in 20 g oleogel) in Sf1, Sf3, and Sf5 promoted fat crystal aggregation. In Sf10, the size of the globular particles was reduced with a concurrent decrease in the density of these structures. Accordingly, we observed that SFL promoted the fat crystal aggregation till Sf5. In Sf10, the size of the globular particles was reduced with a concurrent decrease in the density of these structures. It suggests that adding a higher SFL (Sf10) decreased the fat crystal aggregate formation and their density. Further, it can be observed that, among all SFL-based samples, Sf5 showed the highest cluster forming capability. In Sf1, Sf3, and Sf10, the potential to form aggregates was reduced with an increase in SFL. The reduction in the potential to form aggregates was estimated from the area of the distinct spaces, marked with yellow arrows, around the clusters of the globular structures. Sy1 showed larger globular structures than control, whereas, in Sy3, the size appeared similar to that of the control. In Sy5, the size of the globular structures was the smallest among all the SYL-containing oleogels. A further increment in lecithin content in Sy10 formed globular structures that were the largest of the prepared oleogels. An increase in the SYL content gradually reduced the density of the globular structures in Sy1, Sy3, and Sy5. Interestingly, the density of the globular structures in Sy10 was highly dense and was present as a cluster. Among all SYL-based samples, Sy5 showed the maximum occurrence of distinct spaces in between the globular structures. In our study, SFL and SYL were employed as emulsifiers. Broadly lecithins are a class of a mixture of amphiphilic fatty substances that are extracted from both plants and animals. These emulsifiers have been used extensively for food applications. Further, the fat crystal modification ability of the emulsifiers is dependent on their composition [36]. As the used lecithins are from different sources, the composition of the phospholipids varies significantly. This can explain the differential crystal modification properties of SFL and SYL.

#### 2.2.2. Polarized Light Microscopy

The 10x magnification of polarized light microscopy (PLM) was used to evaluate the crystal morphology of the prepared oleogels (Figure 2). Such a micrograph is relevant for visualizing the micro-architecture of fat products, such as oleogels. The analysis of such micrographs provides relevant information for characterizing various lipid-based products, such as oleogels that exhibit birefringent nature due to the presence of crystallized fats. The bright regions observed in the PLM micrographs can be correlated with the crystalline regions in the samples. This is because the crystalline regions of the fats can diffract the polarized light [37]. Hence, the overall intensity of the image was calculated by using average pixel intensity (API), which can provide information about the overall crystallinity of the oleogel samples (Appendix A) [38]. API can be considered as the quantification strategy of the PLM micrograph. The polarized light micrograph of the control sample suggested a homogeneous distribution of globular fat structures and an API of 35.12. Some bright dots appeared in certain parts of the micrograph, which were reported as grainy structures in the previous work [39]. The appearance of smaller grainy structures is attributed to the interaction between the minor components, such as n-alkanes and nonacosane of CW, with hentriacontane (major alkane). The co-crystallization of hydrocarbons with a fraction of wax esters during the crystallization process can be reasoned for the appearance of such grainy structures [40]. Upon the addition of SFL in Sf1, the overall brightness of the micrograph was reduced significantly. It was observed that in comparison to the control sample Sf1 showed a lower number of globular and grainy fat structures. However, their sizes were similar to that of the control sample. The lower number of individual crystalline points and a considerable number of void spaces (indicated by the pink arrows) confirm the presence of more amorphous regions in Sf1. One possible reason behind the occurrence of small fat crystals can be the occurrence of rapid crystallization, which, in this case, can be confirmed through crystallization study. Rapid crystallization will support less crystal growth. The API of Sf1 was the lowest among all the developed oleogels. A further increase in the SFL content in Sf3 resulted in a marked reduction in the number of grainy structures. The apparent brightness of the Sf3 micrograph was higher than Sf1 but lower than the control. This was also confirmed from the API value. This observation suggested better crystallinity of Sf3 than Sf1. Interestingly in Sf5, there was a drastic increase in the brightness of micrographs. The API of Sf5 was 64.40, which was higher than the control and the highest among all the SFL samples. The micrograph of Sf5 displayed better packing of the globular structures and more number of bigger grainy structures. These results together indicate an overall rise in crystallinity in Sf5. Nevertheless, the apparent size distribution of the grainy structures was quite broad. At the maximum SFL content (in Sf10), the micrograph showed globular fat structure; however, the number of grainy structures was found lowest in this case compared to all the developed oleogels. The overall brightness of the Sf10 micrograph was also found to be lower (API = 24.50) than that of Sf5. The inclusion of SYL in the lowest concentration in Sy1 showed heterogeneously distributed globular matrices and brighter conjoint fat structures with different types of irregular arrangements (indicated by the yellow arrows). The formation of such structures may delay the crystallization process, which can be verified in further studies. The overall brightness (API = 44.68) of the micrograph was observed to be higher than the control. Due to the compactness of this system, there is a possibility of better crystallinity of oleogels in Sy1. In the case of Sy3, the globular structures were homogeneously distributed, giving an overall increased brightness to the micrograph. It is interesting to note that the API of Sy3 (68.07) was the highest among all of the prepared oleogels. The possible reason behind this can be the extent of the compactness of globular structures. There were only a few void spaces in Sy3. However, the number of grainy bright structures was reduced in this case. A further rise in SYL content in Sy5 showed a micrograph with lower brightness (API = 30.33), which was lower than the control. Sy5 showed an increased number of grainy structures over Sy3. Interestingly, there was a decrease in globular structures and a consequent increase in the voids, which can explain a decrement in the crystallinity of Sy5. At the maximum SYL content (in Sy10), the oleogels displayed a rise in the micrograph’s overall brightness (API = 54.29), but it was lower than Sy3. Sy10 showed the largest grainy structures, which were similar to some of the grainy structures in Sf5, among all the SYL-containing oleogels. These grainy structures were distributed in a relatively homogenous manner throughout the sample matrix. The increased size of the grainy structures in Sf5 and Sy10 might be associated with the self-assembling property of lecithin, wherein phospholipids in lecithin get absorbed at the growth sites of the CW crystals [27]. The process of self-assembly promotes the growth of fat crystals. However, the differential effect of lecithin on prepared oleogels might be due to the difference in the source of the two lecithins [41].

### 2.3. FTIR Analysis

FTIR spectroscopy was carried out to understand the molecular arrangement of raw components and formulated oleogels. The plant-derived oil, such as RBO, majorly constitutes triacylglycerols. The di-and monoacylglycerol represents the minor component of RBO. The spectra of RBO (Appendix A) showed a characteristic peak at 3003 cm^−1^, credited to the stretching vibration of C–H present in =C–H [42]. The high relative content of monounsaturated (oleic acid) and PUFA (linoleic acid) in RBO results in the appearance of this peak [43]. Two sharp bands at 2922 and 2854 cm^−1^ corresponded to stretching vibration of –CH_2_ and –CH_3_ of triglycerides, respectively. The C–H bending vibration in –CH_2_ and CH_3_ was indicated by the spectral peak at 1461 cm^−1^ and a smaller peak at 1379 cm^−1^, respectively. The discrete band in the spectra at 722 cm^−1^ was related to the out-of-plane vibration of –HC=CH– [44]. The transmittance peak of acyl-glyceride, i.e., carbonyl group (C=O), was distinctly visible at 1742 cm^−1^ in the spectra of the RBO, which was from the ester group of unsaturated fatty acid [45]. The C–O vibrational stretching of the ester group was represented with a shoulder peak at 1240 cm^−1^ and a sharp peak at 1157 cm^−1^ [44]. The spectra of CW showed transmittance bands at 2922 and 2854 cm^−1^ that corresponded to the stretching vibration of C–H stretching. Further, the C–H rocking vibration, which is usually seen in long-chain alkanes, is represented by a peak at 724 cm^−1^ [46]. Moreover, C–H bending or scissoring is represented by the appearance of a spectral peak at 1461 cm^−1^. The spectral peak at 1175 cm^−1^ can be attributed to the C–C [47]. A sharp peak at 1734 cm^−1^ ascends due to the ester group’s carbonyl formed by the linkage between fatty acid and glycerol [46]. SYL and SFL are a mixture of phospholipids, triglycerides, and non-phospholipid compounds [48]. The FTIR spectra of SFL and SYL are represented in Appendix A. Both spectra displayed a broader peak at ~3300 cm^−1^, indicating the stretching vibration of –OH [49]. The spectral peak of 3010 cm^−1^ corresponded to the bond vibration of alkene. The transmittance peaks of 2922 and 2854 cm^−1^ correspond to asymmetric vibrations in the stretching of –CH_3_ and –CH_2,_ respectively [50]. The scissoring vibration of –CH_2_ and –CH_3_ was depicted at 1465 and 1371 cm^−1^. A distinct peak at 722 cm^−1^ corresponds to the in-plane CH_2_ deformation. The carbonyl C=O stretching vibration of lecithin was confirmed with an appearance of a spectral band at 1742 cm^−1^. The active vibrations of PO_2_^−^ and P–O–C occurred in the spectral region between 1200 and 870 cm^−1^, which showed a peak maxima at 1055 cm^−1^ [50].

Further, the FTIR spectral information was used to understand the interaction of all the raw components within the oleogels (Figure 3). In consideration of the instrument’s spectral resolution, there was no noticeable shifting in the observed peak positions among the spectra of all the samples. The recorded IR spectra revealed a minor peak at 3003 cm^−1^, supposedly from RBO, linked to the =C–H stretching. The presence of this peak in the spectra confirms the high degree of unsaturation in the formulations. Two sharp peaks at 2922 and 2852 cm^−1^ are ascribed to stretching in C–H of CH_3_ and CH_2_.

Additionally, the bending vibration of C–H in CH_3_ and CH_2_ was depicted through spectral peaks at 1463 and 1379 cm^−1^ [51]. The peaks at 720 cm^−1^ in the spectra of all the formulations corresponded to the bending of (CH_2_)_n_. A sharp band at 1744 cm^−1^ in the FTIR spectra of all formulations linked to the stretching vibration in C= present in the aliphatic ester of RBO and CW. A shoulder peak at 1240 cm^−1^ and a sharp peak at 1163 cm^−1^ represent the stretching vibration of the C–O ester group. FTIR spectra can provide information about the acyl chain packaging. The observed bands for CH_2_ rocking (722 cm^−1^) and CH_2_ bending (1461 cm^−1^) mentioned before are common to orthorhombic subcell packing [52]. No peaks agreeing to O–H stretching in the IR spectrum of the prepared samples were observed, although it was present in both the SFL and SYL. This might be due to the low content of lecithin in the oleogels. It can be concluded that the samples either lack or have almost little hydrogen bonding among the components.

### 2.4. Thermal Studies

#### 2.4.1. Gelation Kinetics

The gelation kinetics was studied to understand the crystallization mechanism of SFL and SYL added RBO-CW oleogels (Appendix A). According to [53], the crystallization process in lipid-based materials consists of three distinct phases: (a) nucleation phase, (b) growth phase, and (c) consolidation phase. The sharp decline of temperature in the temperature vs. time graphs corresponds to the nucleation phase, whereas the curves and constant regions represent the growth and consolidation phases, respectively (Appendix A) [54]. In control, at 672 s, a curving was observed (marked by a green arrow) that indicates the initiation of secondary crystallization (Table 2). The time associated with the curving region can be linked to the beginning of secondary crystallization. SFL in Sf1 slightly reduced the nucleation phase and displayed the onset occurred at 663 s. A subsequent increase in SFL content in Sf3 and Sf5 further reduced the onset of secondary crystallization to 652 and 636 s, respectively. However, in Sf10, this increases to 665 s, which is very close to Sf1. These observations suggested that the inclusion of SFL resulted in the early start of secondary crystallization compared to the control. This might be due to the loose molecular packing in SFL-containing samples. This may further suggest these samples’ good spreadability, which can be confirmed in the mechanical study [40]. The addition of SYL in Sy1 also resulted in the early onset of secondary crystallization (604 s) than that of the control. Upon further increment in SYL content in Sy3 and Sy5, there was a rise in the onset of secondary crystallization to 671 and 720 s, respectively. This indicated that in Sy3 and Sy5, the introduction of SYL either did not have any effect or hindered the onset of secondary crystallization, respectively. The onset of secondary crystallization drastically reduced in Sy10 to 609 s, which was lower than the control. The distinct gelation kinetics of the SYL-containing oleogel was due to the different organization of fat structures. Okuro et al. (2018) have reported that the crystallization kinetics get affected based upon the content of lecithin in the formulations [27]. The consolidation phase in the crystallization process corresponds to thermal equilibrium (indicated by the red arrow) in fat components [53]. In control, the equilibrium of the crystallization process was achieved at 1932 s. The addition of SFL in Sf1 and Sf3 reduced the time to attain the equilibrium stage to 1701 and 1776 s, respectively. However, in Sf5 and Sf10 the equilibrium was achieved at 1934 and 1941 s, respectively. This indicates that Sf5 and Sf10 took relatively more time in comparison to Sf1 to attain complete crystallization. The introduction of SYL in Sy1 resulted in the longest time (2021 s) to attain the equilibrium stage among all the prepared oleogels. The possible reason behind this can be the formation of conjoint fat structures visible in the micrograph of Sy1. Furthermore, this delay may promote crystal growth in Sy1. On further increment in the SYL in Sy3 and Sy10, a delay in the attainment of equilibrium was observed as compared to the control. The Sy5 was the fastest (1680 s) to attain the equilibrium stage. The presence of irregular-sized grainy structures as displayed in the micrograph of Sy5 can be due to the quicker attainment of the equilibrium stage. This may lead to the formation of loosely packed structures that may affect the mechanical property of the sample.

For further analysis, the initial portion (0–150 s) of the crystallization kinetics profile was modeled by using an exponential decay function to find out the rate of crystallization (k) (Equation (4)).
(1)$$x\left(t\right)=a\times {\;e}^{-\mathrm{kt}}$$
where “a” represents initial temperature (in our case 50 °C), “t” is time in second, and “k” represents crystallization rate in °C/s. The rate of crystallization of the control was found to be 3.6 °C/ms (Table 2). Sf1 showed rapid crystallization with a “k” value of 5.2 °C/ms. This might be one of the reasons for low crystal growth that resulted in smaller-sized grainy structures in Sf1. An increment in SFL in Sf3 and Sf5 resulted in a lower crystallization rate, i.e., 3.9 and 3.4 °C/ms, respectively. However, in Sf10, it was increased again to 4.6 °C/ms. The delay in attainment of equilibrium with a lower rate of crystallization might be one of the reasons for the appearance of few bigger grainy structures in Sf5. The low crystallization rate also suggested the presence of fewer crystal defects and better mechanical properties of Sf5. The inclusion of SYL in Sy1 showed a crystallization rate of 4.0 °C/ms, which was the least among the SYL-added oleogels. The delayed attainment of equilibrium can be allied with the development of stable and compact fat structures observed in the PLM micrograph of Sy1. The “k” values obtained in Sy3, Sy5, and Sy10 were 6.7, 5.6, and 5.1 °C/ms, respectively. Sy3 showed the highest rate of crystallization among all prepared samples. Although Sy3 showed improved crystallinity, which is evident from the brightness of the PLM micrograph, the high rate of crystallization may have resulted in the formation of fat crystals with defects [55].

#### 2.4.2. DSC Analysis

The thermal properties of the prepared oleogels were found out by using the differential scanning calorimeter (DSC) in the temperature range from 0 to 100 °C. Any physical changes in the oleogels during the melting (endotherm) and cooling (exotherm) processes can be related to the changes in the polymorphic transition. The thermograms representing the melting and cooling of the prepared oleogels are presented in Appendix A. The endotherms of all the samples displayed two peaks, marked in green and black arrows. The first broad peak (green) appeared at a lower temperature than the second, which was relatively a sharper peak (black). One possible reason behind these two peaks’ appearance is the influence of CW, which is composed of multiple components [56]. The other possibility can be the presence of two different fat structures, i.e., globular and grainy structures, which were visualized in PLM micrographs. The broader peak that occurred at a lower temperature was deconvoluted by using Origin Pro software to gain information regarding the peak position and area of the peaks (Table 3). As discussed before, the key components of CW are n-alkanes. An n-alkane, e.g., n-hentriacontane with 31 carbon (C31), is present in abundance in CW [57]. It has been reported that pristine C31 exhibits three endothermic transitions. These peaks were associated with the transitions of orthorhombic to monoclinic phase, monoclinic to rotator phase, and rotator to liquid phase [58]. The T_m_ of crude CW was reported as 54.40 °C in the previous work. However, a lowering in T_m_ has been proposed in the case of multicomponent systems [59]. The control sample displayed a broader band, which peaks at 26.66 (T_m1_), and a second sharp endothermic peak that is positioned at 40.37 °C (T_m2_). The T_m1_ was attributed to the melting of fatty acids and fatty alcohols. However, the higher T_m2_ corresponds to hydrocarbons and wax esters of CW, which are mid-melting point compounds [40]. The T_m_ at ~40 °C is attributed to the monoclinic to the rotator phase transition of hydrocarbon crystals [58]. On the addition of lecithin, both the melting events, i.e., T_m1_ and T_m2_ values, were reduced in all the samples in comparison to the control. The lowering of melting temperatures indicates a reduction in the thermal strength, which may also affect the mechanical properties of the samples. The addition of SFL showed a decrease of T_m1_ till Sf5. A consequent increase in the SFL content in Sf10 increased T_m1_. The lowest T_m1_ in Sf5 might affect the strength of samples but can improvise spreadability [60]. The addition of SYL showed a similar T_m1_ value in Sy3 and Sy5 to that of the control. However, this value was lower in Sy1 and Sy10. The trend of T_m2_ among the SFL samples was observed as Sf3 > Sf5 > Sf1 > Sf10. On inclusion of SYL, the value of T_m2_ was decreased till Sy3, followed by an increase in Sy5. The T_m2_ value in Sy10 decreased again and became similar to Sy1. This indicates a correlation between the thermal and mechanical strength of Sy1 and Sy10. The other minor n-alkanes (nonacosane and tritriacontane) form a molecular packing with CW [61]. The cooling profile in all the samples except Sy5 peaked at ~40 °C (T_c_), indicating the major role of n-alkanes in guiding the crystallization. A similar crystallization peak has been observed for oleogels prepared by using CW in soybean oil [62]. Although few studies have reported the role of vegetable oil in maintaining the melting and crystallization temperature [63], it is quite possible that the wax has more influence on the thermal properties of the oleogels. Interestingly, the addition of lecithin has shown a lowering in the onset crystallization temperature, suggesting that the temperature displacement might be due to the lecithin [27]. The occurrence of rapid cooling (in contrast to gelation kinetics) in DSC can be a reason for getting similar T_c_, irrespective of the lecithin content.

### 2.5. Mechanical Studies

The textural properties of the prepared oleogels were studied by performing the spreadability test (Figure 4a,b). It is crucial to study the properties of solid fats, mainly if they are used as spreads, jams, margarine, and other food products. During the penetration stage, it was observed that the resulted force was increased with a subsequent upsurge in the depth of the probe. At the highest depth, the force attained a maximum value. Consequently, when the probe is pulled back to its initial position, the force instantaneously transforms to negative values. The positive and negative peak of the curve was measured as the firmness (F_0_) and stickiness (S_0_) of the oleogel, respectively. The area under the positive peak of the curve was calculated and marked as “work of shear (C_0_)”, a value that indicates the cohesive energy stored within the sample [64]. While the area under negative peak was measured as “work of adhesion (A_0_)” and indicates the adhesive energy of the sample. The negative (−) sign in the work of adhesion and stickiness is due to the opposite movement of the test probe.

In Table 4, the firmness, work of shear, stickiness, and work of adhesion of the prepared oleogels are presented. The F_0_ value of the control was found to be 594.99 ± 39.17 g. The addition of SFL showed a reduction in the F_0_ value of samples in comparison to the control. In Sf1, a significantly lower F_0_ value than that of the control (*p* < 0.05) was observed. The possible explanation for this is the reduced number of fat structures, which gave an overall higher amorphous region, as observed in the PLM micrograph of Sf1. The increment in the SFL content did not lead to the formation of oleogels with firmness higher than control. The high cluster forming capability (observed in surface topography) and a slower rate of crystallization of SF5 can be the reason for its highest F_0_ value, though this rise was not significant [54]. The inclusion of lower content of SYL in Sy1 and Sy3 displayed similar firmness to that of the control (*p* > 0.05). A further rise of SYL in Sy5 showed a drastic reduction in the F_0_ value from Sy1, Sy3, and the control. This makes perfect correlation with the PLM micrograph of Sy5 that displayed less density of globular structures. The increased F_0_ value at the highest SYL content in Sy10 was significant from Sy5. However, their firmness was similar to the control. The higher crystallinity due to larger grainy structures observed from the micrograph of Sy10 might be responsible for the enhancement in firmness value. A similar kind of observation in terms of firmness of oleogel was reported in Reference [25], where the SYL and ethylcellulose were used in high oleic canola oil to prepare the oleogel. In most of the table fats, a high correlation has been observed between spreadability and cohesiveness, whereas a low correlation has been reported between spreadability and adhesiveness [65]. The cohesiveness (C_0_) of the prepared samples is presented in Table 4. The C_0_ of the control was measured to be 3513.43 ± 197.89 g-mm. Although the addition of SFL has shown an overall reduction of mean C_0_ values in all samples compared to the control, the reduced value in the case of Sf1 was only found significant to that of the control (*p* < 0.05). The inclusion of SYL in Sy1 and Sy3 showed similarity to that of the control in regard to cohesiveness. The average C_0_ value of Sy1 was highest among SYL samples, though this rise was insignificant. The compactness of grainy structures, which was evident in the PLM micrograph of Sy1, can be the possible reason for the improved cohesiveness. On further increasing the SYL content, the C_0_ value of Sy5 was drastically reduced from the control, Sy1, and Sy3. The addition of SYL at the highest content showed no effect on the C_0_ of Sf10 from the control. However, the value showed a significant difference from Sy5.

The spreadability can be implicitly measured from both stickiness and firmness [60]. The stickiness (S_0_) of the control was estimated to be −271.06 ± 20.32 g and was highest amongst all the prepared oleogels. Previous studies suggest that too-firm or -sticky gel does not spread easily onto a surface [60]. A consequent decrease in the S_0_ values with an increased SFL content was observed among the SFL-containing samples, though the decline was insignificant. The addition of SYL did not show any variation in the stickiness of Sy1 and Sy3 from the control. However, a sudden drop in the S_0_ value of Sy5 from the control and Sy1 was noteworthy. At the maximum SYL content in Sy10, there was no change in the stickiness from control and the rest of the SYL-containing samples. The adhesiveness (A_0_) of the prepared samples is presented in Table 4. The A_0_ value of the control was calculated as −851.19 ± 31.35 g-mm. The A_0_ values were observed to be reduced in all the SFL samples in comparison to the control. The reduction in the case of Sf5 was only found significant from the control. The addition of SYL in Sy1 and Sy3 showed similar adhesiveness to the control. However, a rise in the content of emulsifiers in Sy5 showed a sudden drop in the A_0_ value from Sy1, Sy3, and the control. At maximum SYL content in Sy10, the A_0_ value appeared similar to the control, but the rise in Sy10 from Sy5 was substantial.

In a nutshell, the two emulsifiers that were used in the study showed few effects on the mechanical properties of the samples. Among the SFL samples, not much variation in the mechanical strength was observed from the control and samples. However, in Sf5, a lower SD from the mean values of mentioned parameters suggested forming repeatable fat structures that make it worth discussing. The SYL samples’ significant highest F_0_ and C_0_ in Sy1 indicated better mechanical strength, molecular interaction, and stability.

## 3. Conclusions

Oleogels that show excellent oil-structuring properties could better replace the traditional trans- and saturated-fat structuring methods [54]. This study evaluated the influence of changing concentrations of lecithin, a food-grade emulsifier, from sunflower and soya sources on the color, crystal morphology, molecular, thermal, and mechanical properties of CW/RBO oleogel. The obtained results suggested that the use of differently sourced lecithin affects the microstructure and textural properties of the oleogels. The colorimetry analysis showed the prepared oleogels appeared light yellowish white with positive b* values. A uniform appearance of globular structure with alteration in size and density was observed in the surface topography of the samples. The PLM microscopy revealed the presence of globular and grainy fat structures, whose size and arrangements were varied depending on the composition of the oleogel. Samples Sf5 sand Sy1 displayed bright micrographs with a dense arrangement of fat structures. The IR spectroscopy of the samples confirmed no significant difference in the molecular level interactions, even in the presence of lecithin. Further, Sf5 and Sy1 showed a lower crystallization rate in their respective samples, indicating their capability to form fat crystals with fewer defects. The DSC endotherm displayed two melting peaks attributed to the melting of fatty acids, fatty alcohols, hydrocarbons, and wax esters. The addition of both the lecithin showed a lowering in the melting temperature of samples. Furthermore, the prepared oleogels resulted in different spreadability test parameters (F_0_, A_0_, S_0_, and C_0_) due to the varying lecithin content. By taking all four parameters into account, Sf5 and Sy1 showed good spreadable properties among all prepared oleogels. From the abovementioned observations, it can be concluded that the inclusion of lecithin from different sources with varying content showed different effects in physicochemical properties of CW-based oleogels.

## 4. Materials and Methods

### 4.1. Materials

RBO (Patanjali brand, Patanjali Ayurveda Ltd., Haridwar, India) was obtained from the local market of Rourkela, India. CW was procured from Nature’s Tattva, India. SFL and SYL were purchased from Urban Platter, Ltd., Maharashtra, India.

### 4.2. Preparation of Oleogels

Initially, screening was performed to find out the Critical Gelling Concentration (CGC) of CW for gelling RBO. For this purpose, the concentration of CW was ranged from 1% to 5% *w*/*w*. The screening was carried out by dissolving CW in RBO at 90 °C for 25 min to dissolve the CW completely. The homogenous solution of CW and RBO was then kept at 4 °C for 90 min to observe the oleogel formation. The gel formation was confirmed by using the inverted tube method, where the flow of the sample was observed under gravity (Appendix A) [66]. It was identified that 5% (*w*/*w*) of CW in RBO caused the formation of the stable oleogel as compared to the lower wax percentages. Once the CGC was identified, the control sample was prepared by mixing CW and RBO in the required amount (Table 1). Separately, two stock solutions of 0.1% (*w*/*w*) of each type of lecithin (SFL and SYL) were prepared in RBO. This was performed to avoid any error as the amount of emulsifiers that need to be added is quite low. An appropriate amount of stock solution was added to RBO to achieve 1, 3, 5, and 10 mg of lecithin in 20 g of the test samples, resulting in the CW: lecithin ratio of 1:0.001, 1:0.003, 1:0.005, and 1:0.010, respectively. After that, CW was added to the RBO-lecithin mixtures. Then, the samples were melted (i.e., at 90 °C for 25 min) and kept under mild agitation (300 rpm), using a magnetic stirrer to obtain a clear solution (~5 to 7 min). The clear oily mixture was then cooled at room temperature for 5 min and kept under refrigeration (4 °C) for gelation [67]. The composition of the different samples prepared from SFL and SYL is shown in Table 5, and the process is depicted in Figure 5.

### 4.3. Colorimetry Studies

The color measurement of the prepared oleogel was performed with an in-house build colorimeter device. A comprehensive explanation of the hardware of the device has been provided [68]. Initially, calibration of the sensor was performed by using black and white standard placards. For conducting this test, each sample was transferred to 35 mm Petri dishes. Then, the images of samples were taken to calculate different color parameters, such as L* (lightness), a* (+a redness and −a greenness), and b* (+ b yellowness and −b blueness). The whiteness index (WI), yellowness index (YI), and absolute color difference (ΔE) were calculated by using the abovementioned color parameter values. For this purpose, the following formulas were used [31,32]:(2)$$\mathrm{WI}=100-\;\sqrt{{\left(100-L^{*}\right)}^{2}+{\left(a^{*}\right)}^{2}+{\left(b^{*}\right)}^{2}}$$
(3)$$\mathrm{YI}=142.86\frac{b^{*}}{L^{*}}$$
(4)$$\Delta E=\;\sqrt{{\left(L_{C}^{*}-L_{x}^{*}\right)}^{2}+{\left(a_{C}^{*}-a_{x}^{*}\right)}^{2}+{\left(b_{C}^{*}-b_{x}^{*}\right)}^{2}}$$
where
$L_{C}^{*}$, $a_{C}^{*}$, $b_{C}^{*}$
are the control sample parameters values. The numerical values obtained for different lecithin-based oleogel were substituted with symbols having the x tag.

### 4.4. Microscopy Studies

#### 4.4.1. Surface Topology

The surface topography of prepared oleogels was performed by using a Stereo Zoom Light Microscope (SM-2TZ, AMscope, Irvine, CA, USA) that was fitted with an external eyepiece lens camera (MD500, AMscope, Irvine, CA, USA). The molten samples were transferred to 35 mm Petri dishes before the experiment and then kept in a thermal cabinet (4 °C) overnight.

#### 4.4.2. Polarized Light Microscopy

The microstructure of the oleogels was observed through the Leica DM 750 (Wetzlar, Germany) microscope that was custom-designed to visualize polarized light micrographs. The microscope was equipped with ICC 50-HD camera for imaging. The crystal network arrangements in the prepared oleogels were visualized.

### 4.5. FTIR Analysis

The raw materials and prepared samples were analyzed by using the FTIR spectrometer (Alpha-E, Bruker, Bremen, Germany) to comprehend the interactions between components of the oleogels. The scanning was conducted in Attenuated Total Reflectance (ATR) mode, using the zinc selenide ATR crystal. The scanning was carried out in the wavenumber range of 4000–500 cm^−1^, at a spectral resolution of 4 cm^−1^. The samples underwent an average of 24 scans during spectral analysis.

### 4.6. Thermal Studies

#### 4.6.1. Gelation Kinetics

The gelation kinetics of the prepared samples were studied by using an in-house developed temperature measuring device, and the methodology was adapted from the previous work [30]. The device comprised temperature sensors and a data-acquisition module (USB-6251, National Instrument, Austin, TX, USA). The variation in the temperature of each sample was monitored by using an in-house custom-designed LabVIEW^®^ program. The molten samples (10 g) were transported to glass bottles and heated at a constant temperature in the water bath (90 °C). Then, the sensors were inserted into the culture bottles and positioned in a refrigerated water bath (5 °C). The change in temperature of the samples was monitored and recorded for 90 min.

#### 4.6.2. DSC Analysis

Each sample’s melting and crystallization profile was studied by using the differential scanning calorimeter (200 F3 DSC Maia, NETZSCH, Selb, Germany). The acquired thermograms were processed by using Proteus thermal analysis software (NETZSCH, Selb, Germany). The test was conducted under the nitrogen environment. Oleogel samples (~15 mg) were placed in hermetically sealed aluminum pans and a pierced lid. A properly sealed and empty aluminum pan remained as the reference sample. Thermal properties of the prepared oleogel were recorded in the temperature range from 0 to 100 °C during the heating cycle and from 100 to 0 °C during the cooling cycle. The samples were kept at isothermal conditions for 5 min: at 0 °C, before the heating cycle, and for 5 min at 100 °C, before the start of the cooling cycle. The thermal scanning rate was 5 °C/min.

### 4.7. Mechanical Studies

The mechanical behavior of the samples was studied by conducting the spreadability test using the Texture analyzer (TA-HD+, Stable Microsystems, Godalming, UK). The texture analyzer was operated by using a load cell of 5 N at room temperature (20 °C). A 45° perspex cone was used to perform this test. The molten formulations were transferred to the lower perspex cones and kept at 4 °C for two hours to undergo gelation. The upper perspex cone was allowed to penetrate the sample placed in a lower cone at a rate of 1 mm/s. The penetration was programmed to stop till a height of 2 mm from the base of the lower cone. Subsequently, the male cone returned to the early position at the same rate.

### 4.8. Statistical Analysis

The experimental data were analyzed in triplicate and expressed as mean ± standard deviation. To check the significant differences at *p* < 0.05 level, we performed a Student’s *t*-test.

## Figures and Tables

**Figure 1 gels-07-00226-f001:**
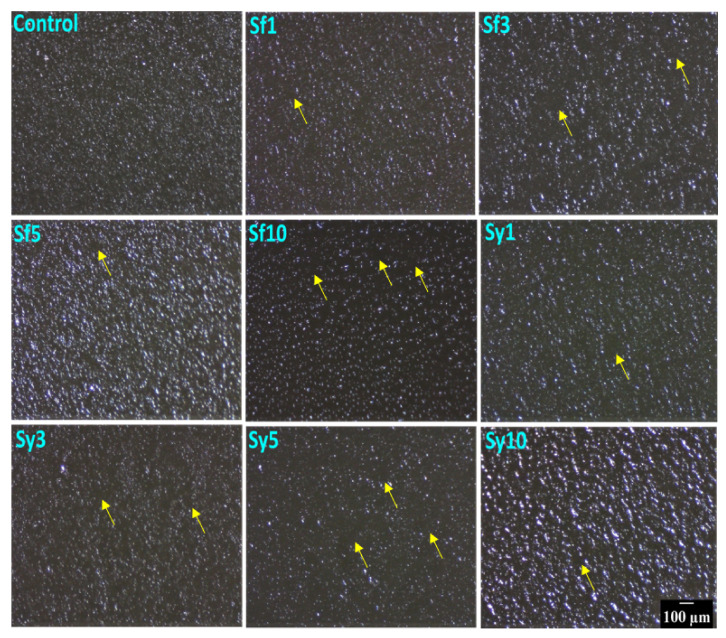
Surface topography images of the prepared samples confirming the presence of globular structures. Scale bar: 100 µm.

**Figure 2 gels-07-00226-f002:**
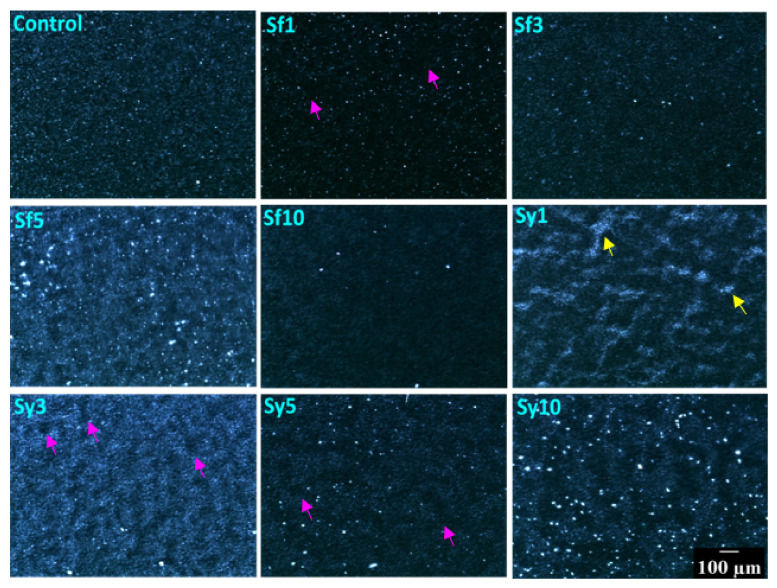
Polarized light micrographs of the prepared samples. Scale bar: 100 µm.

**Figure 3 gels-07-00226-f003:**
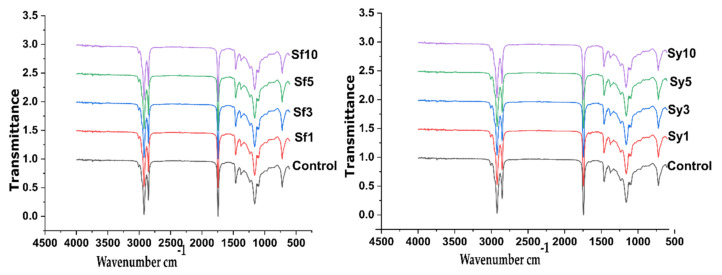
FTIR spectral profiles of the lecithin added samples.

**Figure 4 gels-07-00226-f004:**
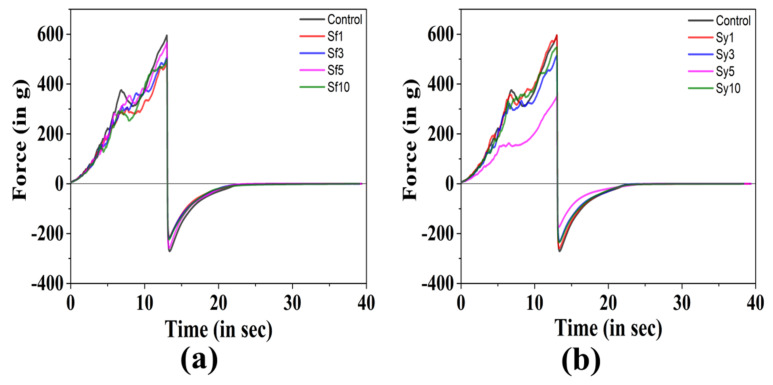
Spreadability profiles of the prepared samples: (**a**) control vs. SFL-containing samples; (**b**) control vs. SYL-containing samples.

**Figure 5 gels-07-00226-f005:**
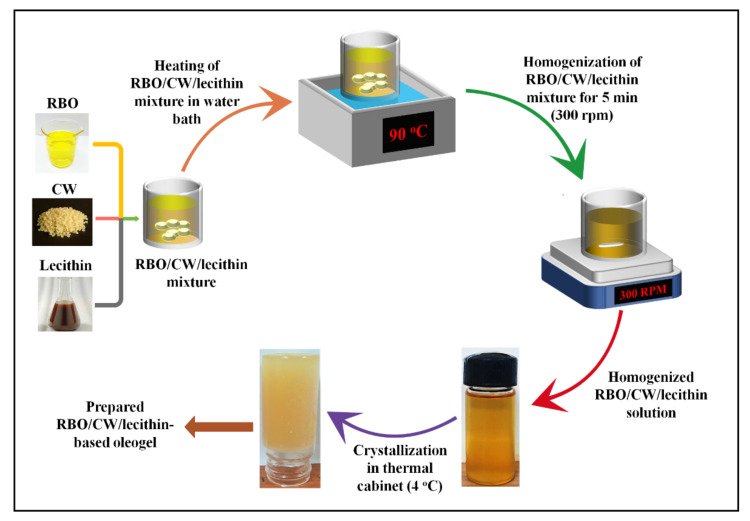
Preparation of lecithin-added oleogels.

**Table 1 gels-07-00226-t001:** Color parameters for the prepared samples. The values are denoted as the mean of the triplicate ± standard deviation.

Parameters						
Samples	L*	a*	b*	YI	WI	ΔE
**Control**	57.45 ± 1.92 ^bcd^	−9.27 ± 3.13 ^afg^	37.15 ± 1.98 ^ef^	92.35 ± 4.49 ^def^	42.69 ± 0.74 ^def^	-
**Sf1**	57.71 ± 1.35 ^abcd^	−10.55 ± 2.03 ^eg^	37.26 ± 0.68 ^de^	92.30 ± 3.18 ^ef^	42.62 ± 1.41 ^ef^	4.23 ± 2.27 ^c^
**Sf3**	52.76 ± 0.85 ^e^	−6.20 ± 1.44 ^abcd^	43.64 ± 3.13 ^a^	118.26 ± 6.55 ^a^	35.38 ± 1.91 ^g^	9.32 ± 1.90 ^a^
**Sf5**	55.89 ± 0.40 ^d^	−4.32 ± 1.17 ^a^	39.61 ± 3.13 ^abcdefgh^	101.18 ± 6.55 ^bcdef^	40.50 ± 1.51 ^f^	6.51 ± 3.93 ^abc^
**Sf10**	57.37 ± 1.54 ^cd^	−7.78 ± 1.23 ^cg^	32.92 ± 1.05 ^gh^	81.97 ± 2.79 ^ghi^	45.56 ± 0.78 ^h^	5.81 ± 1.33 ^abc^
**Sy1**	60.15 ± 1.21 ^abc^	−9.36 ± 0.25 ^dg^	40.02 ± 0.94 ^bce^	95.07 ± 1.89 ^cdef^	42.73 ± 0.82 ^cdef^	5.09 ± 1.42 ^bc^
**Sy3**	59.41 ± 0.36 ^abc^	−5.89 ± 0.74 ^abc^	38.2 ± 1.11 ^de^	91.84 ± 1.45 ^f^	42.93 ± 0.39 ^bcdef^	5.16 ± 1.81 ^abc^
**Sy5**	61.60 ± 0.94 ^a^	−8.83 ± 3.63 ^afg^	34.81 ± 0.46 d^f^	80.72 ± 0.61 ^hi^	47.33 ± 0.51 ^a^	7.20 ± 2.24 ^abc^
**Sy10**	59.93 ± 0.82 ^a^	−7.41 ± 0.35 ^bcfe^	32.67 ± 0.59 ^h^	77.99 ± 2.60 ^i^	47.77 ± 1.14 ^ah^	6.13 ± 1.42 ^abc^

Mean values with different superscripts at the same column are statistically significant (α = 0.05).

**Table 2 gels-07-00226-t002:** Gelation kinetics parameters obtained through exponential decay modeling.

Samples	Temperature vs. Time		Exponential Decay Model
	Onset of Secondary Crystallization (s)	Time to Reach Equilibrium (s)	Initial Rate of Crystallization (k) (°C/ms)
**Control**	672	1932	3.6
**Sf1**	663	1701	5.2
**Sf3**	652	1776	3.9
**Sf5**	636	1934	3.4
**Sf10**	665	1941	4.6
**Sy1**	604	2021	4.0
**Sy3**	671	1981	6.7
**Sy5**	720	1680	5.6
**Sy10**	609	2015	5.1

**Table 3 gels-07-00226-t003:** DSC parameters that were obtained from the deconvoluted endothermic peaks of melting profile and exothermic peaks of crystallization profile.

Samples	Melting				Crystallization	
	Peak (Green Arrow)	Temperature (°C)	Area	Peak (Black Arrow)	Onset (°C)	Peak Temperature (°C)
**Control**	Peak 1	19.7	0.037	40.37	52.13	42.13
	Peak 2	26.66	0.158			
	Peak 3	32.57	0.034			
**Sf1**	Peak 1	23.01	0.72	34.03	50.91	42.16
	Peak 2	22.27	0.40			
	Peak 3	27.85	0.051			
**Sf3**	Peak 1	16.10	0.030	37.82	49.63	42.14
	Peak 2	19.58	0.022			
	Peak 3	26.54	0.24			
**Sf5**	Peak 1	13.24	0.21	35.24	48.39	42.15
	Peak 2	18.51	0.027			
	Peak 3	25.62	0.352			
**Sf10**	Peak 1	20.02	0.458	32.75	48.40	42.16
	Peak 2	25.12	0.018			
**Sy1**	Peak 1	17.68	0.033	35.24	49.64	42.14
	Peak 2	23.11	0.145			
	Peak 3	26.93	0.044			
**Sy3**	Peak 1	19.27	0.209	34.02	48.39	42.14
	Peak 2	26.16	0.092			
	Peak 3	29.00	0.012			
**Sy5**	Peak 1	20.68	0.185	39.08	45.91	39.67
	Peak 2	27.65	0.066			
	Peak 3	32.13	0.019			
**Sy10**	Peak 1	19.00	0.113	35.28	49.67	42.17
	Peak 2	23.12	0.029			
	Peak 3	26.41	0.044			

**Table 4 gels-07-00226-t004:** Different mechanical parameters were obtained from the spreadability test of the prepared samples.

Samples	Firmness (g) (F_0_)	Work of Shear (g-mm) (C_0_)	Stickiness (g) (S_0_)	Work of Adhesion (g-mm) (A_0_)
**Control**	594.99 ± 39.17 ^a^	3512.37 ± 197.89 ^ab^	−271.06 ± 20.32 ^a^	−851.19 ± 31.35 ^a^
**Sf1**	488.19 ± 44.73 ^b^	2951.00 ± 199.43 ^c^	−214.42 ± 36.33 ^ab^	−659.38 ± 103.21 ^abc^
**Sf3**	512.95 ± 55.93 ^ab^	3161.16 ± 359.50 ^abc^	−218.95 ± 35.84 ^ab^	−686.40 ± 125.37 ^abc^
**Sf5**	563.73 ± 32.43 ^ab^	3315.91 ± 55.92 ^abc^	−257.66 ± 19.17 ^a^	−754.97 ± 25.39 ^b^
**Sf10**	510.98 ± 66.01 ^ab^	3093.36 ± 295.55 ^abc^	−225.00 ± 42.34 ^ab^	−721.05 ± 124.14 ^abc^
**Sy1**	598.76 ± 45.12 ^a^	3614.40 ± 203.69 ^a^	−259.14 ± 26.09 ^a^	−819.16 ± 59.80 ^ab^
**Sy3**	525.77 ± 43.70 ^ab^	3121.40 ± 143.46 ^bc^	−241.02 ± 33.08 ^ab^	−742.43 ± 86.72 ^ab^
**Sy5**	357.60 ± 35.28 ^c^	1926.83 ± 212.49 ^d^	−178.38 ± 25.58 ^b^	−525.62 ± 50.66 ^c^
**Sy10**	551.54 ± 62.53 ^ab^	3383.06 ± 466.29 ^abc^	−235.10 ± 44.01 ^ab^	−763.31 ± 72.17 ^ab^

The values in the table are denoted as the mean of the triplicate ± standard deviation (*p* < 0.05). Mean values with different superscripts at the same column are statistically significant (α = 0.05).

**Table 5 gels-07-00226-t005:** Composition of prepared samples.

Samples	CW (in g)	RBO (in g)	Stock (in g)
**Control**	1.0	19.0	-
**Sf1**	1.0	18.0	1.0
**Sf3**	1.0	16.0	3.0
**Sf5**	1.0	14.0	5.0
**Sf10**	1.0	9.0	10.0
**Sy1**	1.0	18.0	1.0
**Sy3**	1.0	16.0	3.0
**Sy5**	1.0	14.0	5.0
**Sy10**	1.0	9.0	10.0

## Data Availability

The data will be available from the corresponding authors on request.

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
