# Peer review of "Variations in Microstructural and Physicochemical Properties of Candelilla Wax/Rice Bran Oil–Derived Oleogels Using Sunflower Lecithin and Soya Lecithin"

_gels, 2021, doi:10.3390/gels7040226_

Round 1

Reviewer 1 Report

To the authors of Variations in Microstructural and Physicochemical Properties of Candelilla Wax-Derived Oleogels Using Sunflower Lecithin and Soya Lecithin manuscript.

My recommendations for you to improve the manuscript are the following:

  1. In the abstract you state that ,,The proposed work is an attempt to 15 understand the impact of differently sourced like sunflower lecithin (SFL) and soya lecithin (SYL) 16 on the various physicochemical properties of CW and rice bran oil (RBO) oleogels,, . In the introduction (lines 95-96) you mention that ,,The present work aims at preparing oleogels by mixing different concentrations of RBO, CW, and lecithin,, This should be improved.
  2. In the introduction, your affirmation lacks support and citations. You need to add citations in lines 38-40 ( for the statement:However, several studies reported  that replacing of saturated fat with unsaturated fat even partially in the daily diet shows  a positive health impact.) Also in lines 87-88 (for the statement: The interaction of lecithin with wax molecules alters the morphology  and dimension of the resultant fat crystals).
  3. You need to improve your writing style. For instance, in lines 48-50, you either write ,,Oleogels contain vegetable oils that are free from  trans fats and contain fewer SFAs than “traditional” solid fats (e.g., butter, tallow, marga-49 rine, etc.) and if they are used as ingredients might improve the nutritional profile of the food products”.
  4. In line 197 please reformulate and explain the phrase,, However, the density of these globular structures was  reduced till Sf3”. In line 204 the same for ,,Accordingly, we observed that SFL promoted the fat crystal aggregation till Sf5”.
  5. In line 220 you mention that ,,In our study, SFL and 220 SYL have been employed as emulsifiers,, while in introduction you affirm that they are used to alter the crystal size. 
  6. Please explain and complete the information about the sample preparation, sinceyou mention that ,, Separately, two stock solutions of 0.1% (w/w) of each type of lecithin (SFL and SYL) were prepared in RBO.,, How was the dissolution achieved? Why did you make this stock solution separately?  Why did you not choose to mix all the  ingredients at the same time? is the 5% Candelilla concentration maintained?

Reviewer 2 Report

This is an interesting research topic, but the manuscript has the following issues:

1.- Lines 53-54. This physical mechanism does not apply to all gelators. According to the ref. 4, it only is for waxes. Please, clarify it or use other general descriptions.

2.- Line 78. The CW chemical description is incomplete, which "hydrocarbons" do you refer here? alkanes, alkynes, alkenes?

3.- The use of RBO should be mentioned in the title, it is very because is part of the innovation you are presenting in this work.  

4.- Please correct the sequence of the sections in the manuscript, first is Material and Methods and then Results and Discussion.

5.- The results presented in Colorimetry Studies have punctual descriptions but lack discussion and/or conclusion about what this characterization contributes to the investigation.

6.- The images in Figure 2 are not there. So, it is not possible to review the contributions in this “Surface topology” section.

7.- Lines 231-233. This sentence is out of context, because the micrographs showed in Fig, 3 do not "provide relevant information for characterizing VARIOUS LIPID-BASED products..."

8.- Line 241. Please, use the correct chemical description to the chemical components mentioned along the manuscript.

9.- The images observed in these micrographs do not correspond to the birefringent aggregates observed in oleogels systems with CW. In all the micrographs I can see that the amount of sample is very thick but the most notable is that the polarization was not used adequately. I do not understand why has used the image editor software if the images given by a PLM are clear, you can see or not birefringent aggregates.

I suggest consulting the following papers for more information about PLM methodology and CW crystal morphology:

Toro-Vazquez, J.F.; Morales-Rueda, J.A.; Dibildox-Alvarado, E.; Charó-Alonso, M.; Alonzo-Macias, M.; González-Chávez, M.M. Thermal and Textural Properties of Organogels Developed by Candelilla Wax in Safflower Oil. J. Am. Oil Chem. Soc. 2007, 84, 989–1000, doi:10.1007/s11746-007-1139-0.

Morales-Rueda, J.A.; Dibildox-Alvarado, E.; Charó-Alonso, M.A.; Weiss, R.G.; Toro-Vazquez, J.F. Thermo-mechanical properties of candelilla wax and dotriacontane organogels in safflower oil. Eur. J. Lipid Sci. Technol. 2009, 111, 207–215, doi:10.1002/ejlt.200810174.

Da Silva T. L., T.; Barrera Arellano, D.; Martini, S. Interactions between candelilla wax and saturated triacylglycerols in oleogels. Food Res. Int. 2019, 121, 900–909, doi:10.1016/j.foodres.2019.01.018.

Chopin-Doroteo, M.; Morales-Rueda, J.A.; Dibildox-Alvarado, E.; Charó-Alonso, M.A.; de la Peña-Gil, A.; Toro-Vazquez, J.F. The Effect of Shearing in the Thermo-mechanical Properties of Candelilla Wax and Candelilla Wax-Tripalmitin Organogels. Food Biophys. 2011, 6, 359–376, doi:10.1007/s11483-011-9212-5.

10.- Suggestion: For truly observe the crystal morphology you need to perform the same thermal profile used in DSC analysis for developing the sample in the glass slides.

11.- Figure 4. It is an odyssey to see the IR spectras, they are very small figures.

12.- FTIR results. All the information presented in this section is valid for the material characterization. However, the section has a very poor conclusion for an Infrared Spectrometry analysis. According to the results presented it appears that there was no interaction between components, so for what you presented all the descriptions? At least explain it to justify this determination in your manuscript.

On the other hand, it is very difficult to have high spectral resolution using ATR system to analyze oleogels with low gelator concentrations. I recommend using FTIR microscope with a heating/cooling stage to control the temperature at which you need to develop the oleogel. In this step, you will observe the vibration bands' characteristics of interactions that you want to explain if they exist.

13.- Figure 5. Again, it is almost impossible to see the changes and transitions discussed in these small figures.

14.- With DSC you can have enough information on the thermal profile and crystallization kinetics of the oleogels which was the objective to perform thermal analyses with the other equipment? It was not discussed. It would be interesting to do the same determination (kinetics of crystallization) using the DSC and the other device manufactured by the colleagues to compare the results and validate.

15.- Section 4.2, Preparation of Oleogels. As you mentioned as justification in the Introduction section, you are investigating for the first time the CGC of CW in RBO, so you must demonstrate the complete experimental design for this (if not in here, describe it in the supplementary material) and establish the most precisely the CGC (only one point in the range of concentrations studied!).

16.- Line 560. What was the objective to gel the solution at 4°C to establish the CGC? A lot of energy is spent. You should revise the methodologies reported to establish CGC.

Suggestion: Aguilar-Zárate, M.; De La Peña-Gil, A.; Álvarez-Mitre, F. de M.; Charó-Alonso, M.A.; Toro-Vazquez, J.F. Vegetable and Mineral Oil Organogels Based on Monoglyceride and Lecithin Mixtures. Food Biophys. 2019, 14, 326–345, doi:10.1007/s11483-019-09583-1.

17.- Line 565. What was the total amount of solution prepared?

18.- Line 567-568. Here, it is imperative to mention the total ratio of the solution components (CW and lecithin).

19.- Line 571. Which temperature was "in refrigeration"?

20.- Table 1. The data shown in this Table is very confusing and does not correspond to the description in the text (lines 556-572). Because you are affirming that the CGC was between 1% to 5%, and never explain why the control was 0.2% of CW in RBO, only as an example, because you omitted a lot of important information of the methodology. It is evident the lack of experimental design.

21.- Line 591. What type of microscope is this? Microscope of light, confocal, polarized, etc... Please, specify this.

22.- Lines 599-600. The methodology is like a list of instructions, so the "attempts" are not the correct expression to clearly indicate the steps to follow.

23.- Revise English grammar and orthography. There are words that are not well written along the manuscript.

24.- Are the words "female" and "male" the correct expression in this methodological context? I suggest using the terms "upper" or "lower" perspex cone to refer to the geometries.

For the above reasons, this manuscript cannot be considered for publication.

Reviewer 3 Report

The paper of Sahu et al. reported the application of Candelilla Wax (CW) for the formation of oleogels, together with rice bran oil and two different sources of lecithin, namely from soya and sunflower. The produced oleogels were then characterized in their microstructural and physicochemical properties.

The work reports several information and different formulations, nevertheless it is opinion of this referee that the manuscript presents clarity issues, requiring major revision and different amendments to be suitable for publication in Gels. Taking in account the Journal level, the average quality of the manuscript have to be enhanced, even concerning the not fulfilled MDPI guidelines.

Some comments are reported hereafter:

Reference to figures and tables in the text should be made as “Figure X” and “Table X”, as reported in the MDPI guidelines.

Line 16: typo “sourced”.

Line 42: It is recommendable to cite some references for the mentioned sources of USFAs. In that way, any interested reader would have a quick way to deepen the fats composition of each matrix. In addition, USFAs can comprehend also PUFAs (Poly-unsatrated fatty acids). Actually, an interesting source for this type of compounds are microalgae, with different composition spectra according to the selected extraction method. This referee suggest to mention this matrix as USFAs’ source. Couple of possible references could be the following:

  • Mercer, P.; Armenta, R.E., Developments in oil extraction from microalgae. J. Lipid Sci. Technol. 2011, 113 (5), 539−547.
  • Tommasi, E.; Cravotto, G.; Galletti, P.; Grillo, G.; Mazzotti, M.; Sacchetti, G.; Samorì, C.; Tabasso, S.; Tacchini, M.; Tagliavini, E., Enhanced and Selective Lipid Extraction from the Microalga tricornutum by Dimethyl Carbonate and Supercritical CO2 Using Deep Eutectic Solvents and Microwaves as Pretreatment, ACS Sutainable Chem. Eng., 2017, 5, 8316-8322.

Figure 1 a), b), c) and d): Statistical significance should be reported in the charts as well, for sake of readability. Furthermore, authors should better specify how they calculated and expressed the statistical significance. As example, Control and Sf1 appear to have overlapping L* values, if considering the S.D.. Furthermore, authors should better explain to readers how the colour evaluation can help to characterize the produced oleogels, i.e. the exploitation in food context.

Figure 2: Figure is missing. For this reason, paragraph 2.2.1 couldn’t be properly evaluated.

Line 92: Concerning the features of lecithin for the formation of organogel, this referee suggest to mention the following work:

  • Shchipunov, Y.A., Lecithin organogel—a micellar system with unique properties. Colloids Surf A , 2001, 183(185), 541–54.

Line 115: For sake of clarity, authors should refer to the experimental paragraph where the preparation of the control sample is described.

Line 237: It is opinion of this referee that a chart (i.e. histograms) of API values (reported in Table S1) could be valuable for the discussion, helping the reader to follow the text.

Line 266: The word “improvisation” it is probably misplaced in the context.

Table 3: The column “initial temperature” is redundant, being this parameter constant for every runs. It is advisable to erase this column adding the information in the caption.

Line 386: The equation miss of an identification number, applied then in the text during the discussion.

Figure 5: It is opinion of this referee that these graphs are not fundamental in the core of the manuscript, having the authors reported an efficient summary in Table 3. Figure 5 can be moved in the Supporting Information, simplifying the paper.

Line 574: Wrong paragraph number.

Table 1: Please, homogenize the capital letters of samples’ label: in the text they are reported differently.

Line 422: an appropriate reference is needed, concerning the n-hentriacontane content in CW.

Line 425: missing subscript for Tm.

Figure 6: It is opinion of this referee that the reported graphs are not fundamental in the core of the manuscript. This is mainly due to the small scale adopted, which make peaks not very appreciable. Being DSC graphs a large number, the best solution is to switch Figure 6 with Table S2 in the supporting information. Hence, values reported in the table would be more useful for readers in the paper, helping the comparison between each sample.

Line 456-458: Sentence should be rephrased due to typo or misformulation.

Line 588: Last sentence should be rephrased due to typo or misformulation.

Line 592/593/603/613/621/632: It is not clear what authors mean with “make”. If they refer to the instruments’ company, it is opinion that the term is not properly used and it can be omitted for simplicity.

Line 614: Authors should report information concerning the developer company of LabVIEW program.

Line 615: Typo: constant.

Line 622: Authors should report information concerning the developer company of Proteus software.

Conclusions: a dedicated sentence should be reported concerning the general difference (if present) between the two lecithin sources, being this topic one of the scopes of the manuscript.

Bibliography:

- Ref 35 and 37 are the same.

- Ref 26 and 41 are the same.

Authors should refer to the MDPI authors guidelines to properly report references. As example:

  1. Author 1, A.B.; Author 2, C.D. Title of the article. Abbreviated Journal Name Year, Volume, page range.

Round 2

Reviewer 1 Report

If the research aims to reveal the influence of different originated lechitin on the oleogels, in the introduction the composition of the lecithin should be described not only qualitatively but also quantitatively in order to highlight the difference  between the soy lecithin and sunflower lechitin. 

For the polarised  light micrographs please mention the magnification.

In line 541-543 it is stated that the firmness of the oleogels is reduced in the samples containing sunflower lecithine and that one possible cause is ,,the reduced number of fat structures, which gave an overall higher amorphous region, observed in the PLM micrograph of Sf1 ,, Your supposition it is not highly scientifically motivated since the candelilla and rice bran oil concentrations are maintained and lecithine acts as crystal modifier. However, your affirmation from the abstract  it better explaining the phenomenon ,,structural reorganization and crystal growth due to
the addition of lecithin affected its mechanical property significantly,, There are not less fat structures  but because of larger crystals more amorphous and less dense network is exhibited.

 Also you should change the phrase ,,On further increment in SFL content till Sf10, the firmness remained the the same for all the samples to that of control,, into ,,The increment in SFL content did not lead to the formation of oleogels with firmness higher than that of the control,,

Discussion regarding the cooling profiles of oleogels resulting from the DSC experiments, described in lines 512-515, should be extended since even the authors state that literature report that the crystalization kinetics get affected based upon the content of lecithin in the formulations. Please also debate the differences in the T Onset (℃). Can the results be correlated with the results from Gelation Kinetics?

The spectral profiles of the lecithin added samples and the control samples cannot be compared in this form.  They should be overlaid in order to better visualise the differences occuring due to different types of lecythin. Also, in my opinion the curent figure 2 (line 376) should be presented in the supplimentary material and the figure from the supplimentary material in the article, since the discussion is based on it.

Reviewer 2 Report

I cannot revise this manuscript in the actual format because the authors left the document with the Word Change Track tool active and then converted the document to PDF. It is almost impossible to read with fluidity and understand the enhancements the authors made. I reject this manuscript because it does not comply with the specifications for structuring the research paper established by Gels journal.

Author Response

The authors apologize to the reviewer for the confusion. However , we want to clarify here that we have uploaded the word document of the manuscript with the track tool on as per the guidelines provided by the editorial board.

Reviewer 3 Report

Authors properly addressed the main comments of this referee. Nevertheless, it is not possible to detect all the suggested references in the paper, that should be provided before the pubblication.

Author Response

The authors are thankful to the reviewer comments. We have highlighted the suggested references in the reference section for simplicity